# Development and Progression of Non-Alcoholic Fatty Liver Disease: The Role of Advanced Glycation End Products

**DOI:** 10.3390/ijms20205037

**Published:** 2019-10-11

**Authors:** Dinali H. Fernando, Josephine M. Forbes, Peter W. Angus, Chandana B. Herath

**Affiliations:** 1Department of Medicine, The University of Melbourne, Melbourne 3084, Australia; hettiyaf@student.unimelb.edu.au; 2Mater Institute, University of Queensland, Brisbane 4102, Australia; Josephine.Forbes@mater.uq.edu.au; 3Liver transplant unit, Austin Health, Heidelberg 3084, Australia; Peter.Angus@austin.org.au

**Keywords:** advanced glycation end products, hepatic Kuppfer cells, hepatic stellate cells, non-alcoholic fatty liver disease, oxidative stress, receptor for advanced glycation end products

## Abstract

Non-alcoholic fatty liver disease (NAFLD) affects up to 30% of the adult population and is now a major cause of liver disease-related premature illness and deaths in the world. Treatment is largely based on lifestyle modification, which is difficult to achieve in most patients. Progression of simple fatty liver or steatosis to its severe form non-alcoholic steatohepatitis (NASH) and liver fibrosis has been explained by a ‘two-hit hypothesis’. Whilst simple steatosis is considered the first hit, its transformation to NASH may be driven by a second hit. Of several factors that constitute the second hit, advanced glycation end products (AGEs), which are formed when reducing-sugars react with proteins or lipids, have been implicated as major candidates that drive steatosis to NASH via the receptor for AGEs (RAGE). Both endogenous and processed food-derived (exogenous) AGEs can activate RAGE, mainly present on Kupffer cells and hepatic stellate cells, thus propagating NAFLD progression. This review focuses on the pathophysiology of NAFLD with special emphasis on the role of food-derived AGEs in NAFLD progression to NASH and liver fibrosis. Moreover, the effect of dietary manipulation to reduce AGE content in food or the therapies targeting AGE/RAGE pathway on disease progression is also discussed.

## 1. Non-Alcoholic Fatty Liver Disease (NAFLD) and Its Prevalence

Non-alcoholic fatty liver disease (NAFLD) is the most common liver disease in the world with varying prevalence depending on the geographical area, age distribution and presence of other risk factors such as diabetes. For example, the prevalence in the USA is 33.6% among the adult population and 10–20% in children, whereas it is 25% in Europe [1], 25% in Asia [2] and 30% among adults in Australia (GESA, 2012). However, the prevalence of NAFLD in Africa, Middle East and Asia has not been well documented. The main risk factors for developing NAFLD are central obesity, dyslipidaemia, type 2 diabetes and elements of the metabolic syndrome [3,4].

NAFLD progresses to non-alcoholic steatohepatitis (NASH) in 20–30% of cases with its sequelae of liver scarring, cirrhosis and liver cancer. Although a number of risk factors and mechanisms have been identified which are associated with the development of NASH, the precise pathogenesis of the condition remains incompletely understood [4].

NAFLD was first reported by Ludwig et al [5]. Steatosis is typically macrovesicular rather than microvesicular in NAFLD and is not associated with specific inflammation [6]. Macrovesicular steatosis is traditionally described as a hepatocyte containing a single large fat droplet pushing the nucleus to the periphery. However, it is not unusual to observe hepatocytes with multiple small to medium-sized fat droplets in vicinity to those hepatocytes with a single large fat droplet. Because a single large fat droplet is believed to be formed by the fusion of multiple small to medium-sized fat droplets, the term macrovesicular steatosis is generally broadened to include those hepatocytes with small to medium-sized fat droplets. On the other hand, microvesicular steatosis is featured by the accumulation of much smaller uniform minute fat droplets dispersed throughout the hepatocytes. The hepatocyte with microvesicular steatosis has centrally located nucleus and foamy cytoplasm [7]. Early in the disease course, the steatosis is most prominent in zone 3, which is the area of liver lobule around the central vein. However, with progression of disease or severity, the steatosis may spread evenly throughout the hepatic acini or become irregularly distributed [8]. The histological features of fatty liver disease are similar regardless of the causative agents, which include NAFLD, heavy alcohol consumption and certain medications. Histological evaluation and pathological assessment remain the gold standard for diagnosis of NAFLD, despite the limitations of the liver biopsies due to sampling errors and differences in the left and right lobes [9,10]. In 2002, the NASH clinical research network (NASH CRN) designed a NAFLD activity score (NAS) which can be used for the full spectrum of NAFLD in three arms; steatosis (0–3), lobular inflammation (0–3), and ballooning (0–2). In the same year, the American association for the study of liver diseases (AASLD) summarized the histopathological abnormalities of NASH. In this report, steatosis, lobular inflammation and hepatocellular ballooning were identified as the necessary components for the diagnosis of NASH. Fibrosis is not necessarily a component for the diagnosis of NASH, although it is usually present. Currently, most pathologists use these criteria, although a complete consensus has not been reached [6].

## 2. Factors That Drive NAFLD Progression

In 1998, Day and James proposed a mechanism to explain why some NAFLD patients never progress to NASH or more severe forms of fatty disease and they called this a ‘two-hit hypothesis’. In this model, they described the deposition of fat in the liver or simple steatosis as the first hit, whereas the progression of steatosis into steatohepatitis requires the involvement of other factors [11]. Consequently, it has been shown that there is a broad range of these factors, including genetic and epigenetic factors, adipokines, the microbiome and dietary factors which can act as the ‘second hit’ (Figure 1) [12]. This review will concentrate on emerging evidence that advanced glycation end products (AGES) and their interaction with the hepatic receptor for AGES (RAGE) are one factor that drives progression from simple NAFLD to NASH and liver fibrosis.

## 3. Advanced Glycation End Products (AGEs): Formation, Metabolism and Their Role in Diseases

Advanced glycation end products are a biologically active group of compounds formed by non-enzymatic reaction between reducing-sugars with proteins, lipids and nucleic acids which were first discovered by Camille Maillard [13]. The structure of AGEs is heterogenous, depending on the reactive groups involved in the reaction. The Maillard reaction begins with reaction of the aldehyde group in a reducing sugar with the amino group in a protein to release a water molecule, producing early glycation products (Schiff base). This reaction is reversible, and the product may undergo hydrolysis again or further rearrange into more stable Amadori products [13]. The rate of rearrangement of Schiff bases to Amadori products varies depending on the protein type. Amadori products may undergo a further series of dehydration and rearrangements, forming AGEs (Figure 2) [13]. This might include both oxidative and non-oxidative pathways, both being irreversible; therefore, AGEs are more stable and accumulate with time.

The rate of formation of AGEs depending on various factors including the concentration of the reactants such as reducing sugars or proteins. It also depends on the lifetime of the protein; proteins with longer half-life tend to glycate more than a protein with a shorter lifetime. The reaction can be catalyzed in the presence of some transitional metal ions [13]. 

The metabolism of AGEs in the body is not fully understood. Despite the bioavailability of orally ingested AGEs being as low as 10% [15], emerging evidence demonstrating the deleterious effects of oral AGEs on health [16], suggests a significant proportion of dietary AGEs are absorbed by the gut.

Studies conducted over past few decades have identified a complex system of specific AGE receptors, which are involved in AGE clearance. It is proposed that macrophages expressing AGE receptors can degrade larger AGE molecules into small soluble AGEs [17] which are then cleared by the kidneys [18] and about 30% of absorbed AGEs are excreted via urine [15]. Liver sinusoidal endothelial cells and Kupffer cells (KCs) also play an important role in clearing AGEs from the circulation [19]. Removal of AGEs by these cells is a slow process that declines with age [20].

The accumulation of AGEs in the body may result in activation of pro-inflammatory and pro-fibrotic pathways which lead to changes in extracellular matrix structure and function. This may be mediated through two pathways; (1) independent of receptors-via forming cross-links between molecules in the basement membrane and (2) interaction with receptors [21].

The receptor independent effects of AGEs on biological systems are mainly due to the formation of crosslinking during the process of AGE formation. Consequently, structural and functional properties of a protein can be significantly altered [13] making them less malleable and more resistant to proteolysis and this may lead to entrapment of proteins, lipoproteins and immunoglobulins [22]. A number of clinical studies have shown that this crosslinking can affect tissue structure and function [23,24,25]. Moreover, it has been shown that AGE adducts residing in vessel walls can directly inactivate nitric oxide (NO)-mediated vasodilatation independent of its receptors [26]. Increased accumulation of AGEs in serum has been linked to the development of nephropathy, retinopathy, and coronary artery disease in patients with type 2 diabetes [27].

Despite many studies carried out to investigate the involvement of AGEs in diseases via receptor-dependent pathways, studies investigating the role of AGEs in liver disease commenced recently [28].

## 4. Advanced Glycation End Products Receptors (RAGEs) and Cell Signaling

To date, there have been at least 8 candidate receptors identified which are capable of binding AGEs. The best characterized receptor among them is known as the receptor for AGEs (RAGE). Other well characterized receptors are the AGE receptors (AGE-R) AGE-R1, AGE-R2, AGE-R3 [29]. A number of studies suggest that the role of these receptors are complex and different receptors might interact or influence each other [30].

RAGE is a 35-kDa, multiligand receptor protein encoded by RAGE gene localized to chromosome 6 [30]. RAGE has been recognized to be the main receptor that binds to AGEs [30]. It has been reported that RAGE is expressed in many cell types, including hepatic Kupffer cells and hepatic stellate cells (HSCs) [31]. RAGE protein is composed of three main regions; (1) an extracellular component made up of a single V-type domain and two C-type domains which are responsible for the ligand binding, (2) a short transmembrane domain and (3) a relatively long (43 amino acids) cytoplasmic tail which is thought to be crucial for most of the intracellular signaling. Alternative mRNA splicing and proteolytic cleavage forms soluble RAGE (sRAGE) and it has been proposed that these soluble isoforms bind to AGEs prior to binding with membrane-bound RAGE and thereby have a protective role.

Binding of RAGE to its ligands produces signals from its cytosolic tail which then leads to activation of many signal transduction pathways resulting in altered gene expression in many different cell types. Activation of signaling cascades includes ERK 1/2 (p42/p44) MAP kinases [32], p38, SAPK/JNK MAP kinases and the JAK/STAT pathway [33], nuclear factor kappa light- chain-enhancer of activated B (NF-κB) [34]. NF-κB activates transcription of a number of genes [30] such as TGF-β1, CTGF [35], VEGF [36], MCP-1 [37], TNF-α and members of the interleukin family [18,34,38].

An important consequence of NF-κB activation is reactive oxygen species (ROS) generation and subsequent upregulation of RAGE itself, setting up a positive feedback loop via binding to the promoter region of the RAGE gene (Figure 3) [39]. A number of in vitro studies have shown that blockade of RAGE can improve mitochondrial damage and reduces oxidative stress induced by AGES [40,41]. Numerous studies have been carried out to demonstrate the involvement of RAGE in a range of diseases. Most of these involvements appeared to be via its cell signaling upon activation, to influence cell survival, cell proliferation, oxidative stress and inflammatory responses [42]. Deleterious effects of RAGE have been studied in diabetes mellitus, atherosclerosis, neurodegenerative disorders (including Alzheimer’s disease), rheumatoid arthritis, chronic renal disease, and inflammatory bowel diseases [30].

Reactive oxygen species generated by NADPH oxidase or NOX activity function as key secondary messengers in numerous downstream signalling pathways in liver inflammation and fibrosis. Studies performed in our laboratory have shown that Kupffer cells that have been exposed to AGEs shown significantly upregulated gene expression of the subunits of NOX enzyme including a cytosolic regulatory subunit p47^phox^ which is required to be translocated to the cell membrane for enzyme assembly and activity, thus acting as a rate-limiting subunit. In addition, AGE exposure also increased Kupffer cell expression of the membrane-bound catalytic subunit of NOX enzyme, gp91phox/NOX-2 (unpublished).

Another important consequence of increased ROS accumulation in tissues is the activation of mammalian target for rapamycin (mTOR). It is known that ROS generation and mTOR activation are tightly linked and that blocking NOX-activated ROS generation could also block mTOR activation [43]. mTOR is a serine/threonine protein kinase involved in various processes including the regulation of cell growth and survival, metabolism and angiogenesis [44]. Recently, it has been suggested that activation of mTOR pathways plays an important role in liver tumorigenesis associated with metabolic syndrome and NASH. Thus, mTOR activation can result in increased cell proliferation and suppression of apoptosis which may contribute to tumorigenesis [45]. This study has provided some compelling evidence that liver tumor formation in mouse models of metabolic syndrome and NASH, and in patients with hepatocellular carcinoma with NASH background is closely linked to the activation of mTOR pathways. However, long-term studies using mTOR inhibitors in these mouse models will be needed to provide further evidence of the relationship between NOX activation and ROS generation and subsequent mTOR activation and development of hepatocellular carcinoma from NASH.

Advanced glycation end products receptor-1 (AGER-1) (OST-48) is a 48 kDa protein and functions as part of the oligosaccharyltransferase complex [46]. Unlike RAGE, AGER-1 is an AGE-specific receptor that appeared to be a negative regulator of the inflammatory response in mesangial cells [47]. 

Advanced glycation end products receptor-2 (AGER-2) was initially referred to as the 80K-H protein (80 kDa) which is expressed in vascular endothelial cells and kidney cells participates in RAGE recognizing complex [29] in diabetic complications and breast cancers [48].

Advanced glycation end products receptor-3 (AGER-3) which is also known as Mac-2 or galectin-3, is a 32 kDa protein shown to play a role in the development of acute inflammation, [49] myofibroblast proliferation and activation [50], cell migration, adhesion, growth and differentiation [51]. Recently it was shown that in diabetes, AGER-3 (galectin-3) contributes to increased insulin resistance by transactivating insulin receptor [52].

## 5. Role of Hepatic Cells in AGE–RAGE Axis-Induced NAFLD Progression

Hepatic stellate cells (HSCs) make up 5–8% of cells in the normal liver and reside in the ‘space of Disse’ in the liver sinusoids [53]. These cells are quiescent under normal physiology, with a cytoplasm containing lipid droplets rich in vitamin A [54]. The role of HSCs is linked primarily to NASH and fibrosis, rather than initial stages of NAFLD. In response to tissue injury, HSCs undergo a dramatic phenotypic change and transdifferentiate into myofibroblast-like cells (activated HSCs) that produce the extracellular matrix (ECM) [55]. Alpha-smooth muscle actin (α-SMA) expression is often used as a marker of HSC activation [56]. Many in vitro studies investigating possible mechanisms of AGE effects have focused on HSCs [28]. The first evidence of the involvement of myofibroblastic HSCs in the rat liver fibrogenesis was shown by Fehrenbach [57]. HSCs express five different types of AGE receptors: Galectin-3, CD36, SR-AI, SR-BI and RAGE and four of them appeared to be up-regulated during HSC activation upon AGE induction [58]. AGEs enhance αSMA expression, profibrotic cytokine TGF-β1 expression and proliferation of HSCs, which is associated with increased ROS generation [39] and autophagy [59]. Increased ROS generation by AGEs-activated HSCs suggests a mechanism for the role for AGEs in NAFLD pathophysiology [60]. Moreover, this might also lead to upregulation of several key signaling pathways as shown by the upregulation of Rac1, PKCδ and p^47^phox [58].

Emerging evidence confirms the hypothesis that Kupffer cells (KCs) are involved in both initiation and progression of NAFLD [61]. In the early stages of the disease, KCs expand rapidly and secrete cytokines and chemokines contributing to a paracrine activation of protective or apoptotic signaling pathways in hepatocytes and the recruitment of other immune cells [62], mediating a protective role. KCs react to gut bacteria-derived lipopolysaccharide (LPS) in an attempt to prevent it entering the peripheral circulation [63]. Toll-like receptor-4 (TLR4) [64] and CD14 [65] recognizes LPS and signals to adaptor proteins leading to the activation of NF-κB and C-Jun N-terminal kinase (JNK) [66]. Furthermore, KCs are involved in free fatty acid (FFA) activated NF-κB pathways [67], resulting in ROS generation/oxidative stress and endoplasmic reticulum (ER) stress [68] which triggers the progression of NAFLD. Rapid release of ROS in response to LPS and other microbial products shifts the balance between antioxidant and peroxide activity in KCs which eventually drives the activation of fibrogenesis [69]. Mechanistic insights into how AGEs contribute to liver injury is provided by studies in isolated KCs which demonstrated that AGEs increase KC proliferation and oxidative stress [60,70], which in turn can activate HSCs.

Hepatocytes exert metabolic and detoxifying functions, initiate the acute phase response and play a central role in NASH patients and in animal models of NASH [71]. AGE formation may result in loss of hepatocyte function, and intracellular accumulation of AGEs have been shown to cause cell death [72]. Moreover, the interaction of extracellular AGEs with its receptor RAGE expressed in hepatocytes [73] promotes inflammation in hepatocytes which is a characteristic feature of NAFLD progression to NASH [74]. Available evidence also suggests that fatty acids are able to stimulate AGE N^ε^-(carboxymethyl)lysine/(CML) accumulation in hepatocytes and subsequently elicits inflammatory reactions via RAGE induction [75].

## 6. Primary Sources of Advanced Glycation End Products

### 6.1. Endogenous AGEs

Advanced glycation end products are formed at a low rate in healthy individuals as a consequence of normal metabolism [76]. However, the rate can be accelerated in certain conditions due to the high availability of reactants, especially sugars, for example, in subjects with diabetes [77]. 

The glycemic index (GI) is the scale that ranks foods according to how much the blood glucose level rises to after food consumption [78] which depends on both the quantity and the quality (proportion of simple and complex carbohydrates) of carbohydrates present in food. Consumption of sugar-sweetened food has increased in the last couple of decades and is positively correlated with metabolic syndrome [79,80], dyslipidemia and insulin resistance [81,82]. Food with high GI can result in elevated circulating glucose, acting as a substantial source of endogenous AGEs [83]. Moreover, low GI food lower the AGE levels in both animals and humans [84,85,86]. However, more work needs to be performed to confirm the exact association between the intake of high-GI foods and the accumulation of AGEs in the body [16]. 

Apart from the high dietary sugar intake that increases the rate of formation of endogenous AGEs, diabetes can accelerate AGE formation due to hyperglycemia. Elevated glucose level shifts the equilibrium towards the right side facilitating the forward reaction of Schiff’s base formation. A variety of diabetic complication such as atherosclerosis, nephropathy, neuropathy and retinopathy have been shown to be exacerbated due to the involvement of AGEs [13,77].

Oxidative stress can accelerate the glycation reaction [87]. It has been demonstrated that the AGEs can be generated by activating the myeloperoxidase pathway in neutrophils [88]. Importantly, the interaction of AGEs with its receptors can generate further oxidative stress forming a positive loop [89]. A study with healthy, non-diabetic twins showed that the levels of the AGE marker CML were much higher in monozygotic compared with dizygotic twins [90]. This suggests that the formation or accumulation of AGEs is influenced by genetic factors as well.

### 6.2. Exogenous AGEs

Although the formation of AGEs occurs endogenously, they can also be formed during preparation of food. Despite the bioavailability of orally ingested AGEs being as low as 10% [15], emerging evidence demonstrates the deleterious effects of dietary AGEs on health [16], suggesting a significant proportion of dietary AGEs is absorbed by the gut. As mentioned above, the glycation reaction was first observed when reducing sugar is heated in the presence of amino acids causing non-enzymatic glycation, forming a characteristic golden-brown color. 

Inducing the Maillard reaction in food preparation is used to obtain the desired food taste, despite its potentially harmful effects in health. It affects the properties of the food such as color, aroma, delicate flavors, texture and protein functionality. However, if the Maillard reaction is too pronounced, undesired quality changes can occur, producing bitter and burnt flavors [91]. Temperature and time are two key factors that determine the rate and the extent of exogenous AGE formation. Other factors that influence the rate include, type of reactants, pH and the water activity [92,93]. Applying high temperatures, usually above 160 °C, accelerates glycation whereas foods cooked in a watery environment (e.g., boiling and steaming) tend to have low levels of AGEs i.e., the Western diet where dry heat is applied has a large amount of AGEs compared to the Asian diet where slow cooking is involved. The amount of AGEs produced during cooking varies as grilling/barbeque > oven frying > deep frying and broiling > roasting > boiling [94]. Microwaving does not increase AGE content dramatically, as it applies for a short period of time.

An example showing the effects of lowering pH on AGE formation is that marination of food with vinegar prior to processing significantly reduces glycation. Lemon juice has the same effects. Baking with sodium bicarbonate (baking soda) on the other hand, increases the pH of the mixture, enhancing the glycation associated browning [95]. The type of reactants presents in the glycation process also determines the rate. For example, pentose sugars are more reactive than hexoses and monosaccharides are more reactive than disaccharides [96].

## 7. Role of Advanced Glycation End Products in NAFLD Progression

Sebekova and colleagues first showed a positive correlation between circulating CML, a circulating form of AGE, levels and the severity of liver diseases and showed improvement after liver transplantation [97]. This was followed by other studies to investigate the relationship between AGE levels and liver disease that demonstrated similar results [98,99]. 

Animal studies from our laboratory showed that AGEs can augment liver fibrosis. In these studies, it was shown that intraperitoneally injected AGEs augmented hepatic fibrosis in a cholestatic model of bile duct-ligated Sprague Dawley rats [100]. Subsequently, work from our laboratory found that dietary AGEs markedly aggravated NAFLD in C57BL/6 mice fed a high-fat diet [60]. Similar findings were reported in CD-1 mice after 6 weeks of feeding with a high-fat diet containing high AGEs [101]. Another study, which used oral gavage of mice with different doses of CML for 28 days, corroborated the hypothesis that CML induces liver injury [102]. Similarly, oral gavage of rats with CML for 12 weeks, significantly increased protein-bound CML in the liver [59].

Whilst exogenous AGEs appear to worsen NAFLD progression, we studied and compared the effects of endogenous AGEs vs. dietary AGEs. Forty weeks of high-fat feeding in mice caused significant increases in steatosis and mild fibrosis compared to standard chow-fed mice. Inducing diabetes at 15 weeks of high fat feeding caused inflammation and severe liver fibrosis. However, high dietary AGEs prepared by baking the high fat diet at 160 °C for 1 h and feeding mice with this diet caused greater fibrosis, which was significantly higher than that of diabetic mice, as illustrated in Figure 4 This suggests that dietary AGEs may have more impact on worsening liver fibrosis in NAFLD than endogenously derived AGEs in diabetes. This could be because dietary AGEs cause leaky gut, leading to translocation of bacteria and bacterial products [103] which travel to the liver via the portal vein where they interact with Kupffer cells leading to activation of these cells and the release of ROS, proinflammatory cytokines and chemokines. Additionally, it also appears that dietary AGEs alter the microbiome [104] which in turn could be responsible for induction of several different mechanisms, affecting the progression of simple fatty liver to liver fibrosis (Figure 3) [105].

## 8. Targeting the AGE/RAGE Axis in NAFLD Progression to Liver Fibrosis

Currently, there is no medical therapy that can be considered as the standard of care for the treatment of NAFLD/NASH. Lifestyle modification should be stressed as being central to all therapeutic modalities despite the difficulties in implementing it in patients. Studies have shown that moderate amounts of weight loss improve insulin sensitivity. Vitamin E is currently recommended in NASH adults with significant fibrosis but without diabetes or cirrhosis [106]. Studies with ursodeoxycholic acid (UDCA), dipeptidyl peptidase-4 (DPP-4) inhibitors [107], pentoxifylline (PTX), selonsertib, tipelukast, emricasan, adhesion molecule vascular adhesion protein-1 (VAP-1) are currently at an experimental stage [108]. Targeting the microbiome using IMM-24e [109], solithromycin and the TLR4 antagonist JKB-121 have been used in clinical trials without a great deal of promise [66]. Antifibrotic agents such as cenicriviroc (CVC), C–C motif chemokine receptor-2/5 antagonists and galectin-3 antagonists [110] are some other potential drugs whose efficacy is yet to be demonstrated.

Published data from our laboratory and others clearly point to the possibility that the AGE/RAGE axis is an attractive target to interfere with the progression of NAFLD [60]. It is possible to inhibit the AGE/RAGE axis using several strategies. As depicted in Figure 5, dietary restriction to lower AGE content and formation in food would be the first choice. The second option is to use AGE inhibitors that can prevent the formation of endogenous AGEs by trapping dycarbonyl compounds, intermediate products of Maillard reaction [111]. Another possible approach would be to target the AGE/RAGE interaction by using drugs that are known to block AGE/RAGE signaling [112].

### 8.1. Impact of Dietary Modifications on NAFLD Progression

As outlined above, temperature and time are two key factors that determine the amount of AGE formation in food and in our experimental studies we showed that baking the high fat diet for 1 h at 160 °C greatly increased AGE content of the diet (Figure 6A) and subsequent liver injury [60]. Evidence suggests that high dietary AGEs increase circulating AGE levels which eventually respond with a fall in response to low AGE diets [15,77]. A positive relationship between dietary AGE intake and circulating AGE levels is supported by evidence from our laboratory where high dietary AGEs lead to high circulating levels of AGEs in mice with NAFLD (Figure 6B). Evidence of the impact of reducing dietary AGEs on disease pathogenesis comes from studies with animal models that demonstrated reducing AGE intake results in prevention of diabetic nephropathy [113]. These findings are consistent with those in diabetic patients where a low AGE diet for 6 weeks resulted in decreased serum AGE levels which were accompanied by decreased inflammatory markers such as C reactive protein [77]. A low AGE content of the food that can be achieved by cooking the food for a shorter period and at lower temperatures has also been shown to provide benefits in animal models with non-diabetic diseases. For example, a low AGE diet prevented intimal proliferation after arterial injury in apolipoprotein-E knockout mice [114].

It is well documented that the Mediterranean diet has many health benefits compared to the Western diets. Accumulating evidence suggests that the Mediterranean diet could be protective against a range of diseases including atherosclerosis, cancer, diabetes, pulmonary diseases, cognition diseases, obesity and metabolic syndrome [115,116]. The Mediterranean diet is mainly a plant-based diet in which fruits, vegetables and nuts are consumed frequently [117] and it has been documented that most beneficial effects appear to be mediated via anti-inflammatory and anti-oxidant potential of this diet [118]. The Mediterranean diet has been trialed in NAFLD patients in Australia and it was found that even modified traditional Mediterranean diet is still beneficial to these patients [119]. 

In addition to fresh plant produce used in the Mediterranean diet, Mediterranean food preparation involves vinegar-marination prior to processing. Recent work in our laboratory found that marination of food with vinegar (acetic acid) prior to baking at 160 °C for 1 h reduces the formation of AGEs in the high fat diet (Figure 6A). The increased dietary intake of AGEs is expected to elevate circulating AGE levels. Indeed, there were elevated levels of AGEs in the circulation of mice fed the baked diet, which is high in AGEs (see Figure 6A), compared to those in mice fed the high fat diet (Figure 6B). This in turn would be expected to increase liver RAGE expression since RAGE is the most characterized AGE receptor that transduces AGE signal by binding to AGEs [30]. Work in our laboratory thus found that increased circulating AGEs in high AGE-fed mice was closely associated with increased RAGE expression in the liver (Figure 6C), reflecting a positive relationship between circulating AGE levels and liver RAGE expression. Vinegar marination of the high-fat diet prior to baking, which reduces AGE content in the diet (Figure 6A), markedly reduced liver RAGE expression, likely due to reduced levels of AGEs reaching the liver. This would be in keeping with the notion that in the liver, AGEs activate RAGE with subsequent increases in oxidative stress [120], which in turn increases endogenous AGE formation and contributes to a vicious cycle of AGE formation, generation of oxidative stress and further RAGE activation, as depicted in Figure 3.

Indeed, the activation of liver RAGE expression with a diet containing high AGEs (Figure 6), can be linked to a cascade of downstream signaling including oxidative stress (Figure 7A) [60], hepatocyte ballooning and HSC activation [70]. and this is expected to drive NAFLD progression to liver fibrosis. In strong agreement with this, as outlined previously, the progression of NAFLD to liver fibrosis was accelerated in mice fed the baked diet containing high AGEs (Figure 7B) [60]. Importantly, however, when dietary intake of AGEs was reduced by vinegar marination prior to baking the diet there was a profound reduction in the expression of RAGE and its downstream molecules such as pro-inflammatory and pro-fibrotic cytokines in the liver [60], leading to a remarkable improvement in NAFLD progression to liver fibrosis (Figure 7B). The fact that RAGE is the signaling receptor molecule that activates downstream signaling by binding to AGEs is strongly supported by the findings that RAGE knockout (RAGE-KO) mice are protected from the harmful effects of the high AGE-containing diet, thus preventing them from rapid progression of NAFLD to liver fibrosis (Figure 7B). 

### 8.2. AGE Inhibitors

Glycemic control reduces AGE accumulation [121] and hypoglycemic drugs or insulin sensitizers such as metformin and pioglitazone reduce blood glucose levels and AGE formation by lowering blood glucose [122] as well as through their ability to trap reactive carbonyl groups [123]. 

There are a number of inhibitors of the glycation reaction. Aminoguanidine and pyridoxamine are two of the well-studied inhibitors that prevent the conversion of early glycation end products to AGEs. Although their mode of action has been proposed to be different in different studies, clinical trials have been successfully completed using these drugs [124]. Thiamine, benfotiamine [125] and metformin [126] have all been studied in clinical trials. LR-90 inhibits AGE formation [127] and ROS generation [128] and has anti-inflammatory effects [127]. OPB-9195 has different modes of anti-AGE actions including anti-oxidant activity and entrapping or reacting with reactive carbonyl compounds [129]. Moreover, alagebrium, which has been originally proposed to be an AGE cross-link breaker, but appears to have multiple other effects on AGE/RAGE, is a well-tolerated and safe drug that has been successfully used in phase 2 clinical trials for diabetic nephropathy and atherosclerosis [125]. A series of ongoing studies in our laboratory will investigate whether alagebrium is indeed a drug that is able to prevent AGE-RAGE signaling in liver disease.

### 8.3. Inhibitors of RAGE or AGE/RAGE Binding

In addition to the demonstration that RAGE-KO mice are protected from the effects of AGEs [60], studies also showed that the blocking of RAGE or the interaction of AGE/RAGE may be beneficial in attenuating disease progression. It has been shown that soluble isoforms of RAGE (sRAGE) act as a decoy compound by binding to circulating AGEs and thereby reduces AGE binding to RAGE and RAGE-activated cell signaling [130,131]. Thus, sRAGE has protective effects against micro and macrovascular complications in diabetes [114], neointimal formation [132], tumor growth and metastasis [133], colitis [38], wound healing response [134], associated macrophage function [41], neuronal dysfunction [135], and lung inflammation [102]. Moreover, studies have also shown that adenoviral over-expression of esRAGE successfully restored the impaired angiogenic response in diabetic mice [136].

## 9. Conclusions

In conclusion, this review has attempted to provide evidence from published studies by our laboratory and others that the steatotic liver that is exposed to high-circulating AGEs responds with increased oxidative stress and inflammation. It is likely that these changes are closely associated with the activation of Kupffer cells and HSCs. The findings from our laboratory also suggest that dietary AGEs have more potent effects on the progression of NAFLD to liver fibrosis than endogenously derived AGEs in diabetes, possibly due to the crosstalk between dietary AGEs and microbiota. Decreasing dietary AGEs by altering food selection and processing such as vinegar marination as well as selecting appropriate cooking methods appear to offset the possible harmful effects afforded by a high AGE diet. Moreover, the findings in RAGE-KO mice presented in this review indicate that RAGE is responsible for the effects of AGEs in worsening NAFLD. This suggests that inhibitors of AGE/RAGE interaction are worth testing in animal models and potentially in patients with NAFLD/NASH. The findings have broad implications for the way we process foods and the dietary advice given to patients with NAFLD. They also suggest a possible role for therapies targeting the AGE/RAGE pathway in the treatment and prevention of NASH and liver fibrosis.

## Figures and Tables

**Figure 1 ijms-20-05037-f001:**
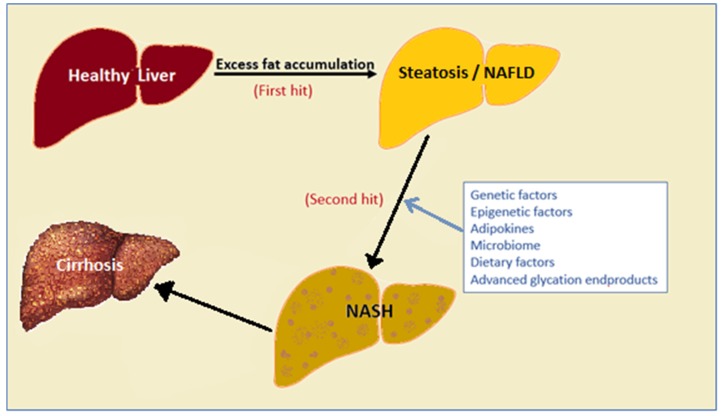
Development and progression of non-alcoholic fatty liver disease (NAFLD) to non-alcoholic steatohepatitis (NASH) and advanced fibrosis. NAFLD can progress into NASH, liver cirrhosis and/or hepatocellular carcinoma due to a number of other factors as depicted in the Figure. NAFLD: non-alcoholic fatty liver disease, NASH: non-alcoholic steatohepatitis.

**Figure 2 ijms-20-05037-f002:**
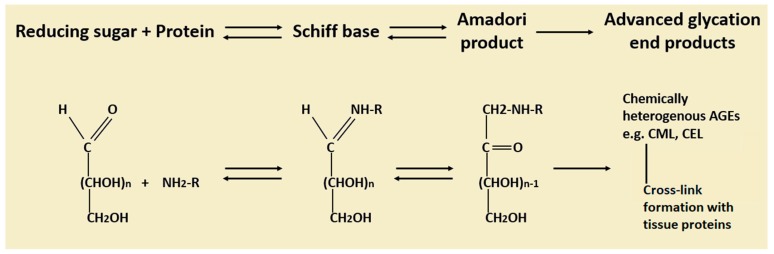
Structure and formation of advanced glycation end products. Formation of advanced glycation end products (AGEs) starts with a reversible reaction between reducing sugars and amino acids resulting in the formation of Schiff bases. Schiff bases then may undergo hydrolysis or rearrange into more stable Amadori products. Amadori products may undergo a further series of dehydration and rearrangements, forming AGEs (adapted from [14]).

**Figure 3 ijms-20-05037-f003:**
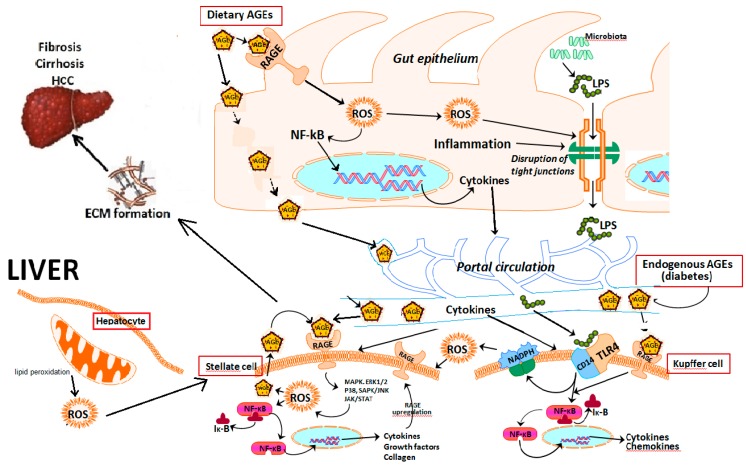
Advanced glycation end products/advanced glycation end products receptor (AGE/RAGE)-induced cellular activation in the liver and gut. Consequences of RAGE activation. Both dietary and endogenous AGEs bind to RAGE expressed by gut and liver cells. RAGE activation causes increased reactive oxygen species (ROS) generation and gene transcription, including that of RAGE itself, thus inducing a positive feedback loop which forms a vicious cycle. In addition, RAGE-induced disruption of tight junctions causes leaky gut allowing gut microbiota-derived endotoxins to enter the portal circulation. Once these endotoxins are bound to receptors present on Kupffer cells, they activate redox sensitive transcription factor NF-κB which in turn leads to increased ROS generation, inflammation, liver fibrosis/cirrhosis and/or hepatocellular carcinoma.NF-κB: nuclear factor kappa B, Iκ-B: inhibitor of nuclear factor kappa B, LPS: lipopolysaccharides (endotoxins), ECM: extracellular matrix, HCC: hepatocellular carcinoma.

**Figure 4 ijms-20-05037-f004:**
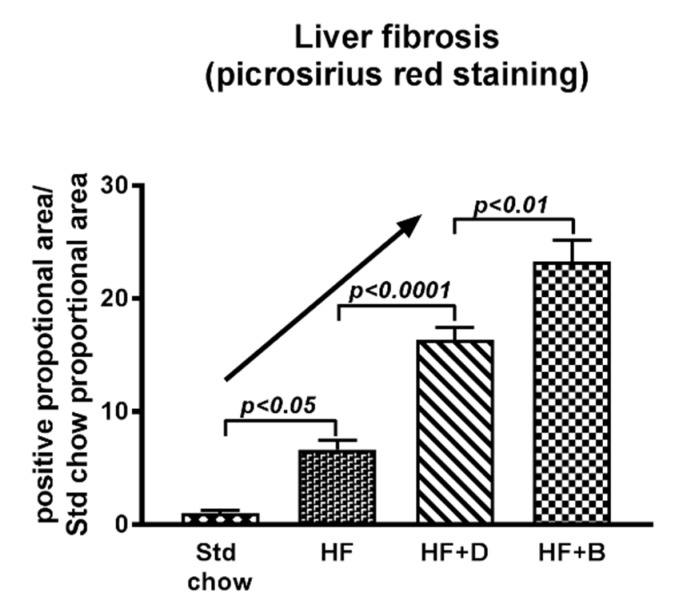
Progression of liver fibrosis during diabetes and high AGE feeding. Liver fibrosis quantified by picrosirius red staining shows that HF diet causes increased fibrosis which is exacerbated in the presence of diabetes. However, high dietary AGEs have worse effect on liver fibrosis. Std chow: standard chow, HF: high fat, D: diabetes, B: baked.

**Figure 5 ijms-20-05037-f005:**
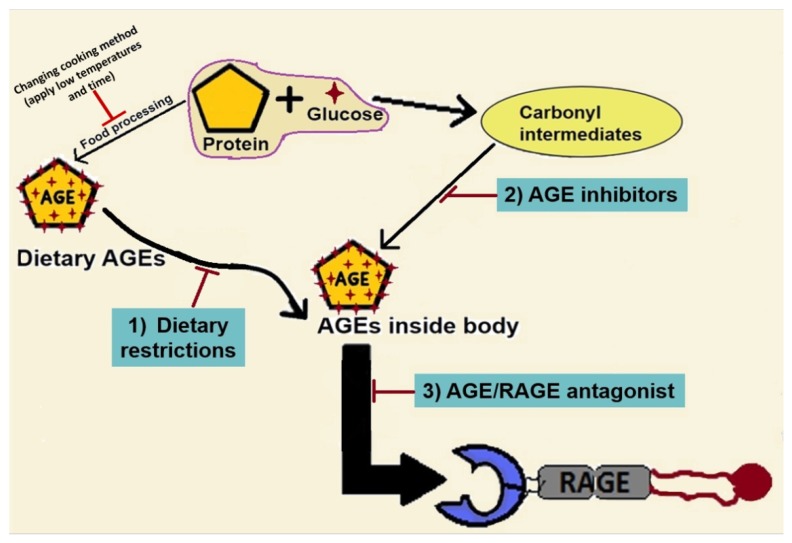
Therapies targeting AGE/RAGE axis. In the treatment and prevention of AGEs-induced NAFLD progression, three strategies could be adapted; (**1**) lowering dietary AGEs and precursors of glycation reaction, (**2**) inhibition of endogenous AGE formation and (**3**) inhibition of AGE/RAGE interaction. AGE: advanced glycation end products, RAGE: receptor for advanced glycation end products.

**Figure 6 ijms-20-05037-f006:**
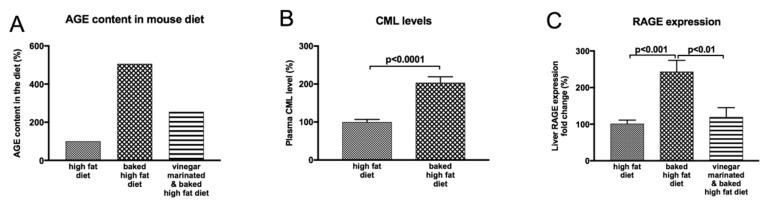
Relationship between dietary AGEs, circulating AGEs and RAGE expression. Baking high fat diet at 160 °C for 1 h dramatically increases AGE content in the diet (**A**), leading to increased circulating AGE (CML) levels (**B**). Moreover, high dietary AGE intake exacerbates liver RAGE expression (**C**). Marination of high-fat diet prior to baking has beneficial effects as reflected by reduced formation of AGEs in the diet (**A**), leading to a reduced liver RAGE expression (**C**). AGE: advanced glycation end products, RAGE: receptor for advanced glycation end products, CML: N^ε^-(carboxymethyl)lysine.

**Figure 7 ijms-20-05037-f007:**
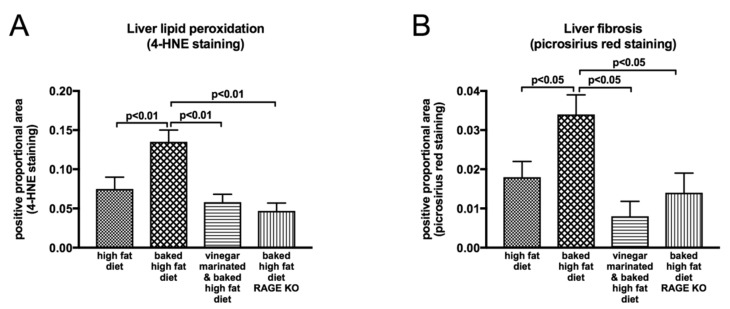
AGE-induced oxidative stress and liver fibrosis. Baking high fat diet at 160 °C for 1 h significantly increases oxidative stress as reflected by increased lipid peroxidation in the liver (**A**) which in turn exacerbates NAFLD progression to NASH and liver fibrosis (**B**). Marination of high fat diet with vinegar prior to baking or RAGE deletion has beneficial effects as reflected by reduced liver oxidative stress (**A**) and liver fibrosis (**B**) [60]. 4-HNE: 4-hydroxynonenal.

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
