# Peer review of "Development and Progression of Non-Alcoholic Fatty Liver Disease: The Role of Advanced Glycation End Products"

_ijms, 2019, doi:10.3390/ijms20205037_

Round 1
Reviewer 1 Report
The current review by Fernando et al. tried to summerize roles of AGEs in progression of NAFLD to NASH and treatment strategies targetting same pathways. The current version of review is interesting but have some major flaws as following:
The abstract is not well written and do not represent the rest the manuscript. And thus should be rewritten. The keywords should be arranged alphabatically. Line 25, NAFLD should be spelled out like the authors did for NASH in line 32. line 36, what did the authors mean by 'steatosis is typically macrovasicular', explain more details about micro or macrovasicular steatosis. Line 37, provide background for zone 3. line 39, give details of various causes. line 49 add refences for the last sentence. Fig 1. after first hit it should be steatosis not NAFLD. Include a paragraph about inflmmation along with ROS for NAFLD progression. There are multiple area of manuscript which sounds repetitive. eg line 128-130 and line 141-143. Except Fig 5 & 6, all figure legeds are indistinguishable from rest of the text. Line 245, title for section 7 is not appropriate and breaking the flow. The authors should incorporate AGE or rage in regards to various hepatic cell types in the title. Line 247, 'Disse'? Line 280, what does the authors mean by 'aggressive histology'? 'Not subjecting to food to high heat (i.e avoiding grilling, ovenfrying, deep frying)" is kind of dieting. Isn't that a lifestyle change? Generally lifestyle change involve healthy habits like dieting and/or excercise. Then how this is better than other lifestyle changing intervention again NAFLD/ NASH? Line 302-317, repetitive with section 5.2. Line 387, explain sRAGE.
Author Response
Point 1: The abstract is not well written and do not represent the rest the manuscript. And thus should be rewritten.
Response 1: We have now rewritten the abstract within the given word limit.
Point 2: The keywords should be arranged alphabatically.
Response 2: Key words now have been reordered alphabetically.
Point 3: NAFLD should be spelled out like the authors did for NASH in line 32
Response 3: NAFLD has now been expanded.
Point 4: line 36, what did the authors mean by 'steatosis is typically macrovasicular', explain more details about micro or macrovasicular steatosis.
Response 4: This has been explained in details in the revised manuscript.
Point 5: Line 37, provide background for zone 3.
Response 5: We have now described what zone 3 is.
Point 6: line 39, give details of various causes.
Response 6: We have now included different causes.
Point 7: line 49 add refences for the last sentence.
Response 7: We now have included a reference for the last sentence.
Point 8: Fig 1. after first hit it should be steatosis not NAFLD.
Response 8: We now have added steatosis into Fig 1.
Point 9: Include a paragraph about inflammation along with ROS for NAFLD progression.
Response 9: We have now included a paragraph about inflammation.
Point 10: There are multiple area of manuscript which sounds repetitive. eg line 128-130 and line 141-143.
Response 10: We have now carefully gone through the paper and removed possible repetitions.
Point 11: Except Fig 5 & 6, all figure legends are indistinguishable from rest of the text.
Response 11: We have now formatted the figure legends.
Point 12: Line 245, title for section 7 is not appropriate and breaking the flow. The authors should incorporate AGE or rage in regards to various hepatic cell types in the title.
Response 12: We have now moved the section 7 up (new section No. 5) to keep the flow appropriately.
Point 13: Line 247, 'Disse'?
Response 13: We have now explained what ‘Disse’ is.
Point 14: Line 280, what does the authors mean by 'aggressive histology'?
Response 14: We mean to say significant fibrosis and have changed this accordingly in the revised manuscript.
Point 15: Not subjecting to food to high heat (i.e avoiding grilling, ovenfrying, deep frying)" is kind of dieting. Isn't that a lifestyle change? Generally lifestyle change involve healthy habits like dieting and/or excercise. Then how this is better than other lifestyle changing intervention again NAFLD/ NASH?
Response 15: We agree with the reviewer that lifestyle changes include dieting and/or exercise. Although dieting may include the use of reduced quantities of desired food, the manipulation of temperature and/or time results in an altered composition of the processed food which are an essential part of Western diet. Therefore, this can be achieved by cooking food at low temperature for a short time, resulting in a reduced formation of AGEs in food. Therefore, other than dieting by means of reduced quantity of food, adjusting cooking/processing practices for popular food can be readily achievable for patients with NAFLD.
Point 16: Line 302-317, repetitive with section 5.2.
Response 16: We have now removed repetitions.
Point 17: Line 387, explain sRAGE.
Response 17: We have now expanded what sRAGE is.
Reviewer 2 Report
The present review paper is well written, the concept is good and authors presented recent information from the literature and from their studies. In general this review will be important for understanding the role of advanced glycation end products in development and progression of NAFLD. There several points to be addressed.
Comments
It would be better to present the picture of AGE with structural formula, showing glycation, glycoxidation of proteins and cross-links formation. In the section “Endogenous AGE” it is advisable to discuss the different types of endogenous AGE, where they are formed (tissue, cellular, molecular level). It is advisable to discuss the role of AGE in NAFLD/NASH-associated liver cancer. Authors show NFkB and MAP kinases as the central molecules in signaling pathways for AGE action. Recently mTOR pathway also has been shown to participate in diabetic NASH and associated hepatocarcinogenesis (Okuno et al, 2018). From several studies, there is a possibility that not the NF-κB but rather mTOR pathway is more important. Please provide more information on signaling pathways. In the chapter “Role of hepatic cells in NAFLD” it is advisable to change the subtitle. Authors discussed only HSCs and KCs, but not other hepatic cells. There is no information on hepatocytes. Does it mean that hepatocytes do not participate and are not affected by AGE? More information is necessary. Figure 4. AGE formation also could be affected by changing food processing technique (prevention of AGE formation), such as changing the time and temperature of cooking (this could be shown in the picture). Please provide information on the type (background) of mice used in the experiments with AGE. Possibly the background differed from experiment to experiment? To know the background of animals is very important in the analysis. Please avoid repetitions in the text.
Author Response
Point 1: It would be better to present the picture of AGE with structural formula, showing glycation, glycoxidation of proteins and cross-links formation.
Response 1: We have included a figure (Fig 2.) to show the steps of AGE formation.
Point 2: In the section “Endogenous AGE” it is advisable to discuss the different types of endogenous AGE, where they are formed (tissue, cellular, molecular level).
Response 2: AGEs are mainly and increasingly formed in the bloodstream, especially when sugar levels are high, for example, in conditions such as diabetes. They can also be formed in many tissues, depending on protein turnover and the availability of reducing sugars in the tissue environment. In this review, we attempted to highlight the relationship between AGEs/RAGE in NAFLD progression with a special emphasis on how dietary AGEs play a leading role in this process. Therefore, we did not intend to discuss in details the sources of endogenous AGEs.
Point 3: It is advisable to discuss the role of AGE in NAFLD/NASH-associated liver cancer. Authors show NFkB and MAP kinases as the central molecules in signaling pathways for AGE action. Recently mTOR pathway also has been shown to participate in diabetic NASH and associated hepatocarcinogenesis (Okuno et al, 2018). From several studies, there is a possibility that not the NF-κB but rather mTOR pathway is more important. Please provide more information on signaling pathways.
Response 3: We have now discussed the mTOR pathway in the manuscript.
Point 4: In the chapter “Role of hepatic cells in NAFLD” it is advisable to change the subtitle.
Response 4: We have now changed it as ”Role of hepatic cells in AGE-RAGE axis-induced NAFLD progression”, and moved the section up as “Section 5”, as suggested by reviewer #1.
Point 5: Authors discussed only HSCs and KCs, but not other hepatic cells. There is no information on hepatocytes. Does it mean that hepatocytes do not participate and are not affected by AGE? More information is necessary.
Response 5: We have now included hepatocytes in this section.
Point 6: Figure 4. AGE formation also could be affected by changing food processing technique (prevention of AGE formation), such as changing the time and temperature of cooking (this could be shown in the picture).
Response 6: We have now adjusted the Figure to show that by changing cooking method with a reduced cooking time can reduce AGE formation. Please note that the new Figure number is Fig 5.
Point 7: Please provide information on the type (background) of mice used in the experiments with AGE. Possibly the background differed from experiment to experiment? To know the background of animals is very important in the analysis.
Response 7: We have now mentioned the background of rats and mice used in our experiments.
Point 8: Please avoid repetitions in the text
Response 8: We have now carefully gone through the paper and removed repetitions.
Round 2
Reviewer 1 Report
The current version is highly improved specially with addition of new figures and modification of texts.